# PrimeDesign software for rapid and simplified design of prime editing guide RNAs

Jonathan Y. Hsu [1,2,3], Julian Grünewald [2,3,4], Regan Szalay [2,3], Justine Shih[2,3], Andrew V. Anzalone[5,6,7], Kin Chung Lam[2,3,4], Max W. Shen [5,6,7,8], Karl Petri [2,3,4], David R. Liu[5,6,7], J. Keith Joung [2,3,4✉] & Luca Pinello [2,4,9✉]

Prime editing (PE) is a versatile genome editing technology, but design of the required guide RNAs is more complex than for standard CRISPR-based nucleases or base editors. Here we describe PrimeDesign, a user-friendly, end-to-end web application and command-line tool for the design of PE experiments. PrimeDesign can be used for single and combination editing applications, as well as genome-wide and saturation mutagenesis screens. Using PrimeDesign, we construct PrimeVar, a comprehensive and searchable database that includes candidate prime editing guide RNA (pegRNA) and nicking sgRNA (ngRNA) combinations for installing or correcting >68,500 pathogenic human genetic variants from the ClinVar database. Finally, we use PrimeDesign to design pegRNAs/ngRNAs to install a variety of human pathogenic variants in human cells.

---

[1] Department of Biological Engineering, Massachusetts Institute of Technology, Cambridge, MA, USA. [2] Molecular Pathology Unit, Massachusetts General Hospital, Charlestown, MA, USA. [3] Center for Cancer Research and Center for Computational and Integrative Biology, Massachusetts General Hospital, Charlestown, MA, USA. [4] Department of Pathology, Harvard Medical School, Boston, MA, USA. [5] Merkin Institute of Transformative Technologies in Healthcare, Broad Institute of Harvard and MIT, Cambridge, MA, USA. [6] Department of Chemistry and Chemical Biology, Harvard University, Cambridge, MA, USA. [7] Howard Hughes Medical Institute, Harvard University, Cambridge, MA, USA. [8] Computational and Systems Biology Program, Massachusetts Institute of Technology, Cambridge, MA, USA. [9] Broad Institute of Harvard and MIT, Cambridge, MA, USA. ✉email: jjoung@mgh.harvard.edu; lpinello@mgh.harvard.edu

Prime editing is a recently developed class of mammalian cell genome editing technology that enables unprecedented precision in the installation of specific substitutions, insertions, and deletions into the genome[1], offering greater versatility than CRISPR nucleases[2–4] and base editors[5,6]. The most efficient prime editing system described to date (referred to as PE3) consists of three components: a fusion protein of a CRISPR-Cas9 nickase and an engineered reverse transcriptase (RT), a prime editing guide RNA (pegRNA), and a nicking sgRNA (ngRNA) (Supp. Fig. 1). The pegRNA targets the Cas9 nickase-RT fusion to a specific genomic locus, but also hybridizes to the nicked single-stranded DNA non-target strand (NTS) within the Cas9-induced R-loop, and serves as a template for reverse transcription to create the "flap" that mediates induction of precise genetic changes (Supp. Fig. 1a–c). The ngRNA directs the Cas9 nickase-RT fusion to nick the strand opposite the flap and thereby biases repair towards the desired change encoded in the flap (Supp. Fig. 1d, e). The complexity of the PE3 system makes it time-consuming to manually design the required pegRNA and ngRNA components. Beyond the need to design the spacer for both guide RNAs, there are multiple other parameters that must be accounted for that can impact prime editing efficiencies, including: primer binding site (PBS) length, reverse transcription template (RTT) length, and distance between the pegRNA and ngRNA target sites.

Here we present PrimeDesign, a user-friendly web application (http://primedesign.pinellolab.org/) (Fig. 1) and command-line tool (https://github.com/pinellolab/PrimeDesign) that automates and thereby simplifies the design of pegRNAs and ngRNAs for single edits, combination edits, and genome-wide and saturation mutagenesis screens. We utilize PrimeDesign to construct PrimeVar, a comprehensive database of candidate prime editing guide RNA (pegRNA) and nicking sgRNA (ngRNA) combinations for installing or correcting >68,500 pathogenic human genetic variants in the ClinVar database. Lastly, we demonstrate the activity of pegRNA and ngRNA designs recommended by PrimeDesign through the installation of human pathogenic variants in human cells.

## Results

**PrimeDesign features**. PrimeDesign uses a single input that encodes both the original reference and the desired edited sequences (Fig. 1a and Supp. Note 1), recommends a candidate pegRNA and ngRNA combination to install the edit of interest (Fig. 1b, Supp. Fig. 2, and Supp. Note 2), provides sequence visualization of the prime editing event and predicted pegRNA secondary structures (Fig. 1c), and enumerates all possible pegRNA spacers, pegRNA extensions, and ngRNAs within optimized parameter ranges (previously defined by the Liu group[1]) for installing the desired edit (Fig. 1d). PrimeDesign enables users to rank pegRNAs based on their predicted specificity (CFD score[7]), provides important annotations for pegRNA (e.g. PAM disruption) and ngRNA (e.g. PE3b) designs, and streamlines the incorporation of PAM-disrupting silent mutations to improve editing efficiency and product purity (Supp. Note 3). In addition, PrimeDesign enables the pooled design of pegRNA and ngRNA combinations for genome-wide and saturation mutagenesis screens (http://primedesign.pinellolab.org/pooled), and ranks the designs according to best design practices[1]. The saturation mutagenesis feature allows for the introduction of mutations at single-base or single-amino acid resolution; PrimeDesign automatically constructs all edits within a user-defined sequence range and generates the designs to install these edits (Supp. Note 4).

**PrimeVar database**. To illustrate the utility of PrimeDesign, we took pathogenic human genetic variants from ClinVar[8]

($n = 69,481$) and designed candidate pegRNAs and ngRNAs for the correction of these pathogenic alleles. Of these pathogenic variants, we found that 91.7% are targetable by at least a single pegRNA spacer with a maximum RTT length of 34 nt (Fig. 2a and Supp. Data 1). An average of 3.7 pegRNA spacers were designed per pathogenic variant, representing multiple options for prime editing to correct each variant. Furthermore, 25.9% of targetable pathogenic variants included at least a single pegRNA that disrupts the PAM sequence, which has been associated with improved editing efficiency and product purity. The PE3b strategy (the design of ngRNAs that preferentially nick the non-edited strand after edited strand flap resolution) is viable for 79.5% of targetable variants (59.7% when only considering mismatches in the seed sequence; Fig. 2b). Lastly, 11.9% of targetable pathogenic variants are amenable to both the PAM-disrupting and PE3b seed-mismatched strategies.

To make all of these ClinVar prime editing designs more accessible, we constructed PrimeVar (http://primedesign. pinellolab.org/primevar), a comprehensive and searchable database for pegRNA and ngRNA combinations to install or correct >68,500 pathogenic human genetic variants. Using either the dbSNP reference SNP number (rs#) or ClinVar Variation ID, candidate pegRNAs and ngRNAs are readily available across a range of PBS (10–17 nt) and RTT (10–80 nt) lengths.

**Installation of pathogenic variants in human cells**. Lastly, we tested recommended pegRNA and ngRNA combinations from PrimeDesign to install 20 different human pathogenic variants associated with genetic diseases including hemophilia A, Duchenne muscular dystrophy (DMD), MPS I and II, and Fabry disease in HEK293T cells (Fig. 3a, Supp. Data 2, and Supp. Note 2). We observed installation of the desired edit at mean frequencies of 10% or more for 7 of the 20 (35%) target sites and at mean frequencies of 1–10% for 6 of the 20 (30%) target sites. For a subset of seven of the desired mutations, we designed additional pegRNAs to assess differences between PE3 and PE3b (Fig. 3b). Generally, we observed mixed trends in the frequencies of the desired edit and a modest reduction in byproducts for PE3b relative to PE3. Lastly, we designed a subset of four additional pegRNAs that introduced PAM-disrupting silent mutations (in addition to the target pathogenic variant) and found that these designs resulted in a mean 1.8-fold increase in the frequency of the desired edit (Fig. 3c).

## Discussion

In summary, PrimeDesign is a comprehensive and general method for facile and automated design of pegRNAs and ngRNAs. Our test of pegRNAs and ngRNAs designed by PrimeDesign to create various edits shows that not all designs yield the desired alterations with high frequencies, therefore, users of PrimeDesign may still need to refine pegRNA choices even after testing initial recommendations. Nonetheless, PrimeDesign should greatly simplify the complicated process of designing candidate prime editing components and thereby increase the use of and accessibility to this powerful and important technology[9–11].

## Methods

**Molecular cloning**. We used a PE2 construct that encodes a P2A-eGFP fusion for cotranslational expression of PE2 and enhanced GFP (eGFP) under control of a CMV promoter (pJUL2440; derived from Addgene no. 132775). For the cloning of pegRNAs (Supplementary Data 2), double-stranded DNA fragments for the pegRNA scaffold, spacer, and 3′ extension were formed by annealing oligos with compatible overhangs for ligation. The fragments were then ligated using T4 ligase (NEB) and cloned into the BsaI-digested pUC19-based hU6-pegRNA-gg-acceptor entry vector (Addgene no. 132777). For nicking gRNA (ngRNA) cloning, spacer oligos were duplexed and ligated into the BsmBI-digested pUC19-based

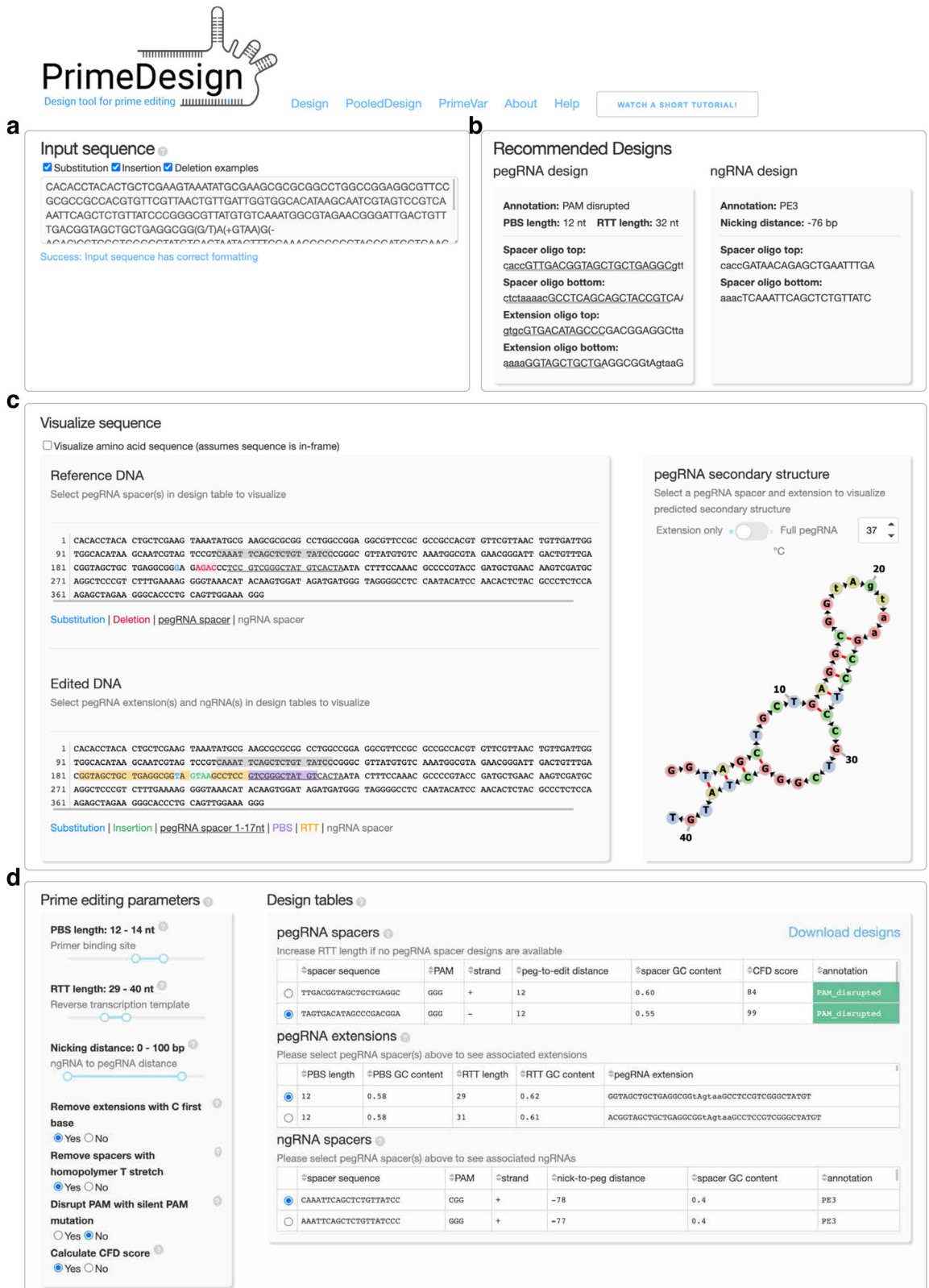

**Fig. 1 PrimeDesign web application. a** PrimeDesign takes a single sequence as input encoding both the original reference and desired edited sequences, **b** recommends a candidate pegRNA and ngRNA combination to install the edit of interest, **c** provides sequence visualizations of the edit of interest, selected pegRNA and ngRNA designs, and predicted pegRNA secondary structures, and **d** enables the interactive design of both pegRNAs and ngRNAs that can be downloaded as a summary table.

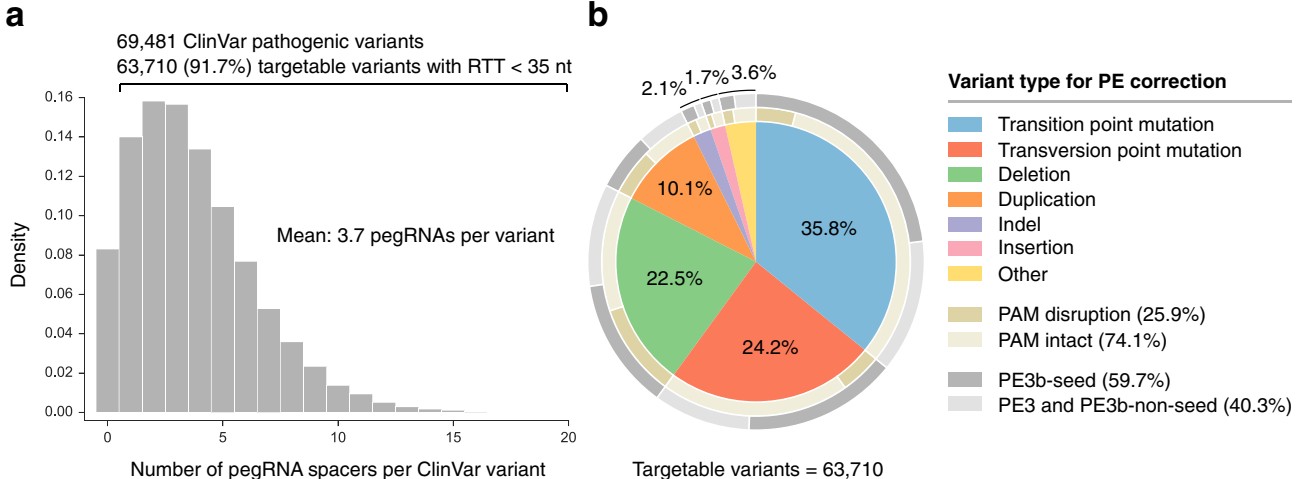

**Fig. 2 PrimeDesign analysis of the ClinVar database. a** The distribution of the number of designed pegRNA spacers per ClinVar variant. Candidate pegRNAs were determined based on the requirement of RTT length <35 nt and the RT extension to have a minimum homology of 5 nt downstream of the edit. **b** The 63,710 (91.7%) targetable ClinVar variants classified by type. The inner ring (gold) represents the proportion of targetable variants by type where at least one pegRNA could be designed to disrupt the PAM sequence (dark gold). The outer ring (gray) represents the proportion of targetable variants by type where at least one ngRNA could be designed for the PE3b strategy where the mismatch lies in the seed sequence (PAM-proximal nucleotides 1–10) (dark gray). See Supplementary Data 1 for details.

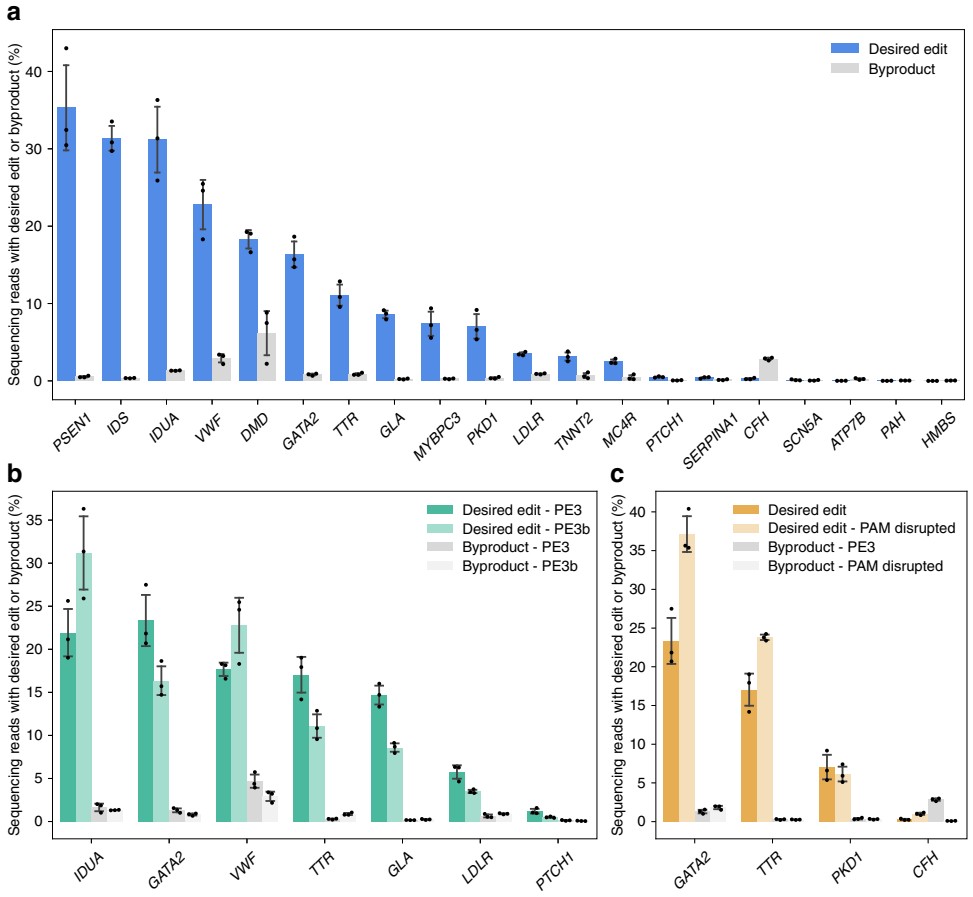

**Fig. 3 Installation of human pathogenic variants in HEK293T cells with PrimeDesign. a** Overview of prime editing efficiencies for the installation of 20 human pathogenic variants in HEK293T cells, using PrimeDesign recommendations. Desired edit refers to sequencing reads containing only the edit of interest, while byproduct refers to sequencing reads containing any mutation(s) outside of only the edit of interest (i.e. indels, desired edit and indels). **b** Comparison between PE3 and PE3b editing strategies. **c** Assessing the effects of PAM-disrupting silent mutations on prime editing efficiencies. Mean ± s.d. of $n = 3$ independent biological replicates. Some of the data shown in **a** are also represented in **b** and **c**. See Supplementary Data 2 for details. Source Data is available in the Source Data file.

hU6-SpCas9 gRNA entry vector BPK1520 (Addgene no. 65777). All pegRNA and ngRNA plasmids were transformed into chemically competent E.coli (XL1-Blue, Agilent). Plasmids used for transfection were midi (PE2) or mini prepped (gRNAs) using the Qiagen midi plus or miniprep kits.

**Cell culture**. STR-authenticated HEK293T cells (CRL-3216) were grown in Dulbecco's modified Eagle medium (DMEM, Gibco) containing 10% fetal bovine serum (FBS, Gibco) and 1% penicillin-streptomycin antibiotic (Gibco). Cells were kept in a 5% $CO_2$ incubator at 37 °C. Cells were passaged every 2–3 days as cells reached 80% confluency. Cells did not exceed passage 13 for all replicates in this experiment. Mycoplasma testing of the cell culture media took place every 4 weeks with the MycoAlert PLUS mycoplasma detection kit (Lonza) and showed negative results for the duration of this study.

**Transfections**. HEK293T cells were seeded into 96-well flat-bottom cell culture plates (Corning) for PE treatment at $1.2 × 10^4$ cells/well. Transfections were carried out 18–24 h post-seeding with 30 ng PE2 plasmid, 10 ng pegRNA, and 3.3 ng ngRNA plasmid per transfection (per well, in a 96-well plate). TransIT-X2 (Mirus) was used as the lipofection reagent at 0.3 μL per transfection.

**DNA extraction**. Post-transfection (72 h), HEK293T cells were washed using 1x PBS (Corning) and lysed with 43.5 μL of gDNA lysis buffer (100 mM Tris, 200 mM NaCl, 5 mM EDTA, 0.05% SDS), 1.25 μL of 1 M DTT (Sigma), and 5.25 μL of Proteinase K per well for 96-well plate experiments. The plates were put into a shaker (500 rpm) at 55 °C overnight, and gDNA was extracted using 1.5x paramagnetic beads. Beads with bound gDNA were washed with 70% ethanol three times using a Biomek FXP Laboratory Automation Workstation (Beckman Coulter) and then eluted in 35 μL 0.1x EB buffer (Qiagen).

**Targeted amplicon sequencing**. The gDNA concentrations of several samples from different pegRNAs/replicates were measured using the Qubit dsDNA HS Assay Kit (Thermo Fisher). The first PCR was performed to amplify the genomic regions of interest (200–250 bp) using 10–20 ng of gDNA. Primers for PCR1 included Illumina-compatible adapter sequences (Supplementary Data 2). A synergy HT microplate reader (BioTek) was then used at 485/528 nm with the Quantifluor dsDNA quantification system (Promega) to measure the concentration of the first PCR products. PCR products from different genomic amplicons were then pooled and cleaned with 0.7x paramagnetic beads. The second PCR was performed to attach unique barcodes to each amplicon using 50–200 ng of the pooled PCR1 products and barcodes that correspond to Illumina TruSeq CD indexes. The PCR2 products were again cleaned with 0.7x paramagnetic beads and measured with the Quantifluor system before final pooling. The final library was sequenced using an Illumina Miseq (Miseq Reagent Kit v.2; 300 cycles, 2 × 150 bp, paired-end). The FASTQ files were downloaded from BaseSpace (Illumina).

**Analysis**. Amplicon sequencing data were analyzed with CRISPResso version 2.0.42 with HDR mode. Downstream analysis was sourced from 'CRISPResso_quantification_of_editing_frequency.txt.' The frequency of *Desired edit* was determined by taking HDR Unmodified and dividing by Reads_aligned_all_amplicons and the frequency of *Byproduct* was determined by taking the sum of HDR Modified, Reference Modified, Ambiguous and dividing by Reads_aligned_all_amplicons.

**PrimeDesign analysis on ClinVar variants**. The ClinVar database was accessed April 8th 2020. Variants were filtered with the following conditions: (1) included a valid GRCh38/hg38 coordinate, (2) labeled as Pathogenic for the column "ClinicalSignificance", and (3) contained a unique identifier determined by the concatenation of columns "Name," "RS# (dbSNP)," and "VariationID." All variants with ambiguous IUPAC code were converted into separate entries with non-ambiguous bases for downstream analysis. Following these steps, the total number of ClinVar variants totaled 69,481. Sequence inputs were formatted for all entries for both the installation and correction of these pathogenic variants. After running PrimeDesign on the ClinVar variants, candidate pegRNA designs were filtered with two criteria: (1) maximum RTT length of 34 nt and (2) minimum homology of 5 nt downstream of the edit. The pegRNAs with PAM disrupted annotations have mutations in the dinucleotide GG of the NGG motif, and the ngRNAs with *PE3b*, *PE3b non-seed*, and *PE3b seed* annotations have mismatches anywhere in the protospacer, mismatches outside of PAM-proximal nucleotides 1–10, or mismatches within PAM-proximal nucleotides 1–10, respectively.

**Construction of PrimeVar database**. The filtered ClinVar variants from the PrimeDesign analysis were used to build a comprehensive database of candidate pegRNA and ngRNA combinations. Prime editing designs are available to install and correct the pathogenic human genetic variants. PrimeDesign was run with a PBS length range of 10–17 nt, RTT length range of 10–80 nt, and ngRNA distance range of 0–100 bp. All of the pegRNA and ngRNA designs for each variant are stored on PrimeVar (http://primedesign.pinellolab.org/primevar).

**Reporting summary**. Further information on research design is available in the Nature Research Reporting Summary linked to this article.

## Data availability
All targeted amplicon sequencing data have been deposited under the BioProject accession number PRJNA688137 at the Sequence Read Archive (SRA), accessible at: https://www.ncbi.nlm.nih.gov/bioproject/PRJNA688137. Information related to the clinical variants in the manuscript are available at: https://www.ncbi.nlm.nih.gov/clinvar/ . Source data are provided with this paper.

## Code availability
PrimeDesign[12] was used to design pegRNAs and ngRNAs (https://github.com/pinellolab/PrimeDesign). Amplicon sequencing data were analyzed with CRISPResso version 2.0.42 with HDR mode (https://github.com/pinellolab/CRISPResso2).

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

## Acknowledgements
L.P. is supported by the National Human Genome Research Institute (NHGRI) Career Development Award (R00HG008399), Genomic Innovator Award (R35HG010717) and CEGS RM1HG009490. J.K.J. is supported by NIH R35 GM118158, NIH RM1 HG009490, the Robert B. Colvin, M.D. Endowed Chair in Pathology, and the Desmond and Ann Heathwood MGH Research Scholar Award. D.R.L. is supported by the Merkin Institute of Transformative Technologies in Healthcare, US NIH grants U01AI142756, RM1HG009490, R01EB022376, and R35GM118062, and the HHMI. A.V.A. acknowledges a Jane Coffin Childs postdoctoral fellowship. J.G. was funded by the Deutsche Forschungsgemeinschaft (DFG, German Research Foundation) – Projektnummer 416375182.

## Author contributions
J.Y.H. developed PrimeDesign. J.Y.H. and J.G. designed the experiments. R.S. and J.S. performed the experiments and analyzed the data. A.V.A., J.G., K.P., and K.C.L provided feedback during the development of PrimeDesign. M.W.S. contributed to the ClinVar analysis. L.P., J.K.J., and D.R.L. supervised the project and provided feedback and guidance. J.Y.H., L.P., J.K.J., and D.R.L. wrote the manuscript with input from all other authors.

## Competing interests
J.K.J. has financial interests in Beam Therapeutics, Chroma Medicine (f/k/a YKY, Inc.), Editas Medicine, Excelsior Genomics, Pairwise Plants, Poseida Therapeutics, SeQure Dx, Inc., Transposagen Biopharmaceutics, and Verve Therapeutics (f/k/a Endcadia). J.K.J.'s

interests were reviewed and are managed by Massachusetts General Hospital and Partners HealthCare in accordance with their conflict of interest policies. J.K.J. is a co-inventor on patents and patent applications that describe various gene editing technologies. D.R.L. is a consultant and co-founder of Prime Medicine, Beam Therapeutics, Pairwise Plants, and Editas Medicine, companies that use genome editing. L.P. has financial interests in Edilytics and SeQure Dx, Inc. L.P.'s interests were reviewed and are managed by Massachusetts General Hospital and Partners HealthCare in accordance with their conflict of interest policies. All other authors declare no competing interests.
