## [Peer Review File · Nature Communications]

PrimeDesign software for rapid and simplified design of prime editing guide RNAs

Jonathan Y. Hsu^{1,2,3,4}, Julian Grünewald^{2,3,4}, Regan Szalay^{2,3,4}, Justine Shih^{2,3,4}, Andrew V. Anzalone^{5,6,7}, Kin Chung Lam^{2,3,4}, Max W. Shen^{5,6,7,8}, Karl Petri^{2,3,4}, David R. Liu^{5,6,7}, J. Keith Joung^{2,3,4*}, Luca Pinello^{2,4,9*}

¹Department of Biological Engineering, Massachusetts Institute of Technology, Cambridge, MA, USA

²Molecular Pathology Unit, Massachusetts General Hospital, Charlestown, MA, USA

³Center for Cancer Research and Center for Computational and Integrative Biology, Massachusetts General Hospital, Charlestown, MA, USA

⁴Department of Pathology, Harvard Medical School, Boston, MA, USA

⁵Merkin Institute of Transformative Technologies in Healthcare, Broad Institute of Harvard and MIT, Cambridge, MA, USA

⁶Department of Chemistry and Chemical Biology, Harvard University, Cambridge, MA, USA

⁷Howard Hughes Medical Institute, Harvard University, Cambridge, MA, USA

⁸Computational and Systems Biology Program, Massachusetts Institute of Technology, Cambridge, MA, USA

⁹Broad Institute of Harvard and MIT, Cambridge, MA, USA

*Correspondence should be addressed to L.P. (lpinello@mgh.harvard.edu) and J.K.J.

(jjoung@mgh.harvard.edu)

Abstract

Prime editing (PE) is a versatile genome editing technology, but design of the required guide RNAs is more complex than for standard CRISPR-based nucleases or base editors. Here we describe PrimeDesign, a user-friendly, end-to-end web application and command-line tool for the design of PE experiments. PrimeDesign can be used for single and combination editing applications, as well as genome-wide and saturation mutagenesis screens. Using PrimeDesign, we construct PrimeVar, a comprehensive and searchable database that includes candidate prime editing guide RNA (pegRNA) and nicking sgRNA (ngRNA) combinations for installing or correcting >68,500 pathogenic human genetic variants from the ClinVar database. Finally, we use PrimeDesign to design pegRNAs/ngRNAs to install a variety of human pathogenic variants in human cells.

Introduction

Prime editing is a new class of mammalian cell genome editing technology that enables unprecedented precision in the installation of specific substitutions, insertions, and deletions into the genome¹, offering greater versatility than CRISPR nucleases^{2,3,4} and base editors^{5,6}. The most efficient prime editing system described to date (referred to as PE3) consists of three components: a fusion protein of a CRISPR-Cas9 nickase and an engineered reverse transcriptase (RT), a prime editing guide RNA (pegRNA), and a nicking sgRNA (ngRNA) (Supp. Fig. 1). The pegRNA targets the Cas9 nickase-RT fusion to a specific genomic locus, but also hybridizes to the nicked single-stranded DNA non-target strand (NTS) within the Cas9-induced R-loop, and serves as a template for reverse transcription to create the “flap” that mediates induction of precise genetic changes (Supp. Fig. 1a-c). The ngRNA directs the Cas9 nickase-RT fusion to nick the target strand (i.e. the strand opposite the flap) and thereby biases repair towards the desired change encoded in the flap (Supp. Fig. 1d-e). The complexity of the PE3 system makes it time-consuming to manually design the required pegRNA and ngRNA components. Beyond the need to design the spacer for both guide RNAs, there are multiple other parameters that must be accounted for that can impact prime editing efficiencies, including: primer binding site (PBS) length, reverse transcription template (RTT) length, and distance between the pegRNA and ngRNA.

Here, we present PrimeDesign, a user-friendly web application

(<http://primedesign.pinelloab.org/>) (Fig. 1) and command-line tool

(<https://github.com/pinelloab/PrimeDesign>) that automates and thereby simplifies the design

of pegRNAs and ngRNAs for single edits, combination edits, and genome-wide and saturation mutagenesis screens. We utilize PrimeDesign to construct PrimeVar, a comprehensive database of candidate prime editing guide RNA (pegRNA) and nicking sgRNA (ngRNA) combinations for installing or correcting >68,500 pathogenic human genetic variants in the ClinVar database. Lastly, we demonstrate the activity of pegRNA and ngRNA designs recommended by PrimeDesign through the installation of human pathogenic variants in human cells.

Results

PrimeDesign features

PrimeDesign uses a single input that encodes both the original reference and the desired edited sequences (Fig. 1a, Supp. Note 1), recommends a candidate pegRNA and ngRNA combination to install the edit of interest (Fig. 1b, Supp. Fig. 2, Supp. Note 2), provides sequence visualization of the prime editing event and predicted pegRNA secondary structures (Fig. 1c), and enumerates all possible pegRNA spacers, pegRNA extensions, and ngRNAs within optimized parameter ranges (previously defined by the Liu group¹) for installing the desired edit (Fig. 1d). PrimeDesign enables users to rank pegRNAs based on their predicted specificity (CFD score⁷), provides important annotations for pegRNA (e.g. PAM disruption) and ngRNA (e.g. PE3b) designs, and streamlines the incorporation of PAM-disrupting silent mutations to improve editing efficiency and product purity (Supp. Note 3). In addition, PrimeDesign enables the pooled design of pegRNA and ngRNA combinations for genome-wide and saturation mutagenesis screens (<http://primedesign.pinellolab.org/pooled>), and ranks the designs according to best design practices¹. The saturation mutagenesis feature allows for the

introduction of mutations at single-base or single-amino acid resolution; PrimeDesign automatically constructs all edits within a user-defined sequence range and generates the designs to install these edits (Supp. Note 4).

PrimeVar database

To illustrate the utility of PrimeDesign, we took pathogenic human genetic variants from ClinVar⁸ (n= 69,481) and designed candidate pegRNAs and ngRNAs for the correction of these pathogenic alleles. Of these pathogenic variants, we found that 91.7% are targetable by at least a single pegRNA spacer with a maximum RTT length of 34 nt (Fig. 2a and Supp. Data 1). An average of 3.7 pegRNA spacers were designed per pathogenic variant, representing multiple options for prime editing to correct each variant. Furthermore, 25.9% of targetable pathogenic variants included at least a single pegRNA that disrupts the PAM sequence, which has been associated with improved editing efficiency and product purity. The PE3b strategy (the design of ngRNAs that preferentially nick the non-edited strand *after* edited strand flap resolution) is viable for 79.5% of targetable variants (59.7% when only considering mismatches in the seed sequence) (Fig. 2b). Lastly, 11.9% of targetable pathogenic variants are amenable to both the PAM-disrupting and PE3b seed-mismatched strategies.

To make all of these ClinVar prime editing designs more accessible, we constructed PrimeVar (<http://primedesign.pinelloab.org/primevar>), a comprehensive and searchable database for pegRNA and ngRNA combinations to install or correct >68,500 pathogenic human genetic variants. Using either the dbSNP reference SNP number (rs#) or ClinVar Variation ID, candidate

pegRNAs and ngRNAs are readily-available across a range of PBS (10-17 nt) and RTT (10-80 nt) lengths.

Installation of pathogenic variants in human cells

Lastly, we tested recommended pegRNA and ngRNA combinations from PrimeDesign to install 20 different human pathogenic variants associated with genetic diseases including hemophilia A, Duchenne muscular dystrophy (DMD), MPS I and II, and Fabry disease in HEK293T cells (Fig. 3a, Supp. Data 2, Supp. Note 2). We observed installation of the desired edit at mean frequencies of 10% or more for 7 of the 20 (35%) target sites and at mean frequencies of 1% - 10% for 6 of the 20 (30%) target sites. For a subset of seven of the desired mutations, we designed additional pegRNAs to assess differences between PE3 and PE3b (Fig. 3b). Generally, we observed mixed trends in the frequencies of the desired edit and a modest reduction in byproducts for PE3b relative to PE3. Lastly, we designed a subset of four additional pegRNAs that introduced PAM-disrupting silent mutations (in addition to the target pathogenic variant) and found that these designs resulted in a mean 1.8-fold increase in the frequency of the desired edit (Fig. 3c).

Discussion

In summary, PrimeDesign is a comprehensive and general method for facile and automated design of pegRNAs and ngRNAs. Our test of pegRNAs and ngRNAs designed by PrimeDesign to create various edits shows that not all designs yield the desired alterations with high frequencies, therefore, users of PrimeDesign may still need to refine pegRNA choices even after

testing initial recommendations. Nonetheless, PrimeDesign should greatly simplify the complicated process of designing candidate prime editing components and thereby increase the use of and accessibility to this powerful and important technology^{9,10,11}.

a

Input sequence

Substitution Insertion Deletion examples

```
CACACCTACACTGCTCGAAGTAAATATGCGAAGCGCGCGGCTGGCCGGAGGGCTTCC
GCGCCCCACAGTGTTCGTAACGTGTTGTTGGTGGCACATAAGCAATCGTAGTCCGTCA
AATTCAGCTCTGTATCCCGGGCGTTATGTGCAATGGCGTAGAACGGGATTGACTGTT
TGACGGTAGCTGCTGAGGCGG(G/T)A(+GTA)G(-
AGACGGCTCGCTCGGCTGCTGACTAATCTTTGCAAGCCCGCTACCGTCTGCTAAG
```

Success: Input sequence has correct formatting

b

Recommended Designs

pegRNA design

Annotation: PAM disrupted
PBS length: 12 nt **RTT length:** 32 nt

Spacer oligo top:
caccGTTGACGGTAGCTGCTGAGGCgtt

Spacer oligo bottom:
ctctaaaacGCCTCAGCAGCTACCGTCA/

Extension oligo top:
gtgcGTGACATAGCCCGACGGAGGctta

Extension oligo bottom:
aaaaGGTAGCTGCTGAGGCGGtAgtaaG

ngRNA design

Annotation: PE3
Nicking distance: -76 bp

Spacer oligo top:
caccGATAACAGAGCTGAATTGA

Spacer oligo bottom:
aaacTCAAATTACAGCTCTGTATC

c

Visualize sequence

Visualize amino acid sequence (assumes sequence is in-frame)

Reference DNA

Select pegRNA spacer(s) in design table to visualize

```
1 CACACCTACA CTGCTCGAAG TAAATATGCG AAGCGCGGG CCTGGCCGGA GGCOTTCCG GCCGCCACGT GTTCGTTAAC TGTGATTGG
91 TGGCACATAA GCAATCGTAG TCCGTCAAAAT TCAGCTCTGT TATCCCGGGC GTTATGTGTC AAATGGCGTA GAACGGGATT GACTGTTTGA
181 CGGTAGCTGC TGAGGCGGT A GAGACCCTCC CTCGGGCTAT GTCACATAA CTTTCCAAAC GCCCCGTACC GATGCTGAAC AAGTCGATGC
271 AGGCTCCCGT CTTTGAAGA GGTAAACAT ACAAGTGGAT AGATGATGG TAGGGCCCTC CAATACATCC AACACTCTAC GCCCTCTCCA
361 AGAGCTAGAA GGGCACCCCTG CAGTTGGAAA GGG
```

Substitution | Deletion | pegRNA spacer | ngRNA spacer

Edited DNA

Select pegRNA extension(s) and ngRNA(s) in design tables to visualize

```
1 CACACCTACA CTGCTCGAAG TAAATATGCG AAGCGCGGG CCTGGCCGGA GGCOTTCCG GCCGCCACGT GTTCGTTAAC TGTGATTGG
91 TGGCACATAA GCAATCGTAG TCCGTCAAAAT TCAGCTCTGT TATCCCGGGC GTTATGTGTC AAATGGCGTA GAACGGGATT GACTGTTTGA
181 CGGTAGCTGC TGAGGCGGT A GAGACCCTCC CTCGGGCTAT GTCACATAA CTTTCCAAAC GCCCCGTACC GATGCTGAAC AAGTCGATGC
271 AGGCTCCCGT CTTTGAAGA GGTAAACAT ACAAGTGGAT AGATGATGG TAGGGCCCTC CAATACATCC AACACTCTAC GCCCTCTCCA
361 AGAGCTAGAA GGGCACCCCTG CAGTTGGAAA GGG
```

Substitution | Insertion | pegRNA spacer_1-17nt | PBS | RTT | ngRNA spacer

pegRNA secondary structure

Select a pegRNA spacer and extension to visualize predicted secondary structure

Extension only Full pegRNA 37

d

Prime editing parameters

PBS length: 12 - 14 nt

Primer binding site

RTT length: 29 - 40 nt

Reverse transcription template

Nicking distance: 0 - 100 bp

ngRNA to pegRNA distance

Remove extensions with C first base

Yes No

Remove spacers with homopolymer T stretch

Yes No

Disrupt PAM with silent PAM mutation

Yes No

Calculate CFD score

Yes No

Design tables

pegRNA spacers

Increase RTT length if no pegRNA spacer designs are available

spacer sequence	PAM	strand	peg-to-edit distance	spacer GC content	CFD score	annotation
<input type="radio"/> TTGACGGTAGCTGCTGAGGC	ggg	+	12	0.60	84	PAM_disrupted
<input checked="" type="radio"/> TAGTGACATAGCCCGCGGA	ggg	-	12	0.55	99	PAM_disrupted

pegRNA extensions

Please select pegRNA spacer(s) above to see associated extensions

PBS length	PBS GC content	RTT length	RTT GC content	pegRNA extension
<input checked="" type="radio"/> 12	0.58	29	0.62	GGTAGCTGCTGAGCCGGtAgtaaGCCCTCCGTCGGGCTATGT
<input type="radio"/> 12	0.58	31	0.61	ACGGTAGCTGCTGAGCCGGtAgtaaGCCCTCCGTCGGGCTATGT

ngRNA spacers

Please select pegRNA spacer(s) above to see associated ngRNAs

spacer sequence	PAM	strand	nick-to-peg distance	spacer GC content	annotation
<input checked="" type="radio"/> CAAATTCAGCTCTGTTATCC	CGG	+	-78	0.4	PE3
<input type="radio"/> AAATTCAGCTCTGTTATCCC	ggg	+	-77	0.4	PE3

Download designs

Figure 1: PrimeDesign web application

a) PrimeDesign takes a single sequence as input encoding both the original reference and desired edited sequences, **b)** recommends a candidate pegRNA and ngRNA combination to install the edit of interest, **c)** provides sequence visualizations of the edit of interest, selected pegRNA and ngRNA designs, and predicted pegRNA secondary structures, and **d)** enables the interactive design of both pegRNAs and ngRNAs that can be downloaded as a summary table.

Figure 2: PrimeDesign analysis of the ClinVar database

a) The distribution of the number of designed pegRNA spacers per ClinVar variant. Candidate pegRNAs were determined based on the requirement of RTT length <35 nt and the RT extension to have a minimum homology of 5 nt downstream of the edit. **b)** The 63,710 (91.7%) targetable ClinVar variants classified by type. The inner ring (gold) represents the proportion of targetable variants by type where at least one pegRNA could be designed to disrupt the PAM sequence (dark gold). The outer ring (grey) represents the proportion of targetable variants by type where at least one ngRNA could be designed for the PE3b strategy where the mismatch lies in the seed sequence (PAM-proximal nucleotides 1-10) (dark grey). See Supplementary Data 1 for details.

Figure 3: Installation of human pathogenic variants in HEK293T cells with PrimeDesign

a) Overview of prime editing efficiencies for the installation of 20 human pathogenic variants in HEK293T cells. Desired edit refers to sequencing reads containing only the edit of interest, while byproduct refers to sequencing reads containing any mutation(s) outside of only the edit of interest (i.e. indels, desired edit and indels). **b)** Comparison between PE3 and PE3b editing strategies. **c)** Assessing the effects of PAM-disrupting silent mutations on prime editing

efficiencies. Mean \pm s.d. of n = 3 independent biological replicates. See Supplementary Data 2 for details.

Acknowledgements

L.P. is supported by the National Human Genome Research Institute (NHGRI) Career Development Award (R00HG008399), Genomic Innovator Award (R35HG010717) and CECS RM1HG009490. J.K.J. is supported by NIH R35 GM118158, NIH RM1 HG009490, the Robert B. Colvin, M.D. Endowed Chair in Pathology, and the Desmond and Ann Heathwood MGH Research Scholar Award. D.R.L. is supported by the Merkin Institute of Transformative Technologies in Healthcare, US NIH grants U01AI142756, RM1HG009490, R01EB022376, and R35GM118062, and the HHMI. A.V.A. acknowledges a Jane Coffin Childs postdoctoral fellowship. J.G. was funded by the Deutsche Forschungsgemeinschaft (DFG, German Research Foundation) – Projektnummer 416375182.

Author Contributions

J.Y.H. developed PrimeDesign. J.Y.H. and J.G. designed the experiments. R.S. and J.S. performed the experiments and analyzed the data. A.V.A., J.G., and K.C.L provided feedback during the development of PrimeDesign. M.W.S. contributed to the ClinVar analysis. L.P., J.K.J., and D.R.L. supervised the project and provided feedback and guidance. J.Y.H., L.P., J.K.J., and D.R.L. wrote the manuscript with input from all other authors.

Competing Interests

J.K.J. has competing interests in Beam Therapeutics, Chroma Medicine (f/k/a YKY, Inc.), Editas Medicine, Excelsior Genomics, Pairwise Plants, Poseida Therapeutics, SeQure Dx, Inc., Transposagen Biopharmaceuticals, and Verve Therapeutics (f/k/a Endcadia). J.K.J.'s interests

were reviewed and are managed by Massachusetts General Hospital and Partners HealthCare in accordance with their conflict of interest policies. J.K.J. is a co-inventor on patents and patent applications that describe various gene editing technologies. D.R.L. is a consultant and co-founder of Prime Medicine, Beam Therapeutics, Pairwise Plants, and Editas Medicine, companies that use genome editing. L.P. has competing interests in Edilytics and SeQure Dx, Inc. L.P.'s interests were reviewed and are managed by Massachusetts General Hospital and Partners HealthCare in accordance with their conflict of interest policies. All other authors declare no competing interests.

Methods

Molecular cloning

We used a PE2 construct that encodes a P2A-eGFP fusion for cotranslational expression of PE2 and enhanced GFP (eGFP) under control of a CMV promoter (pJUL2440; derived from Addgene no. 132775)). For the cloning of pegRNAs (Supplementary Data 2), double-stranded DNA fragments for the pegRNA scaffold, spacer, and 3' extension were formed by annealing oligos with compatible overhangs for ligation. The fragments were then ligated using T4 ligase (NEB) and cloned into the Bsal-digested pUC19-based hU6-pegRNA-gg-acceptor entry vector (Addgene no. 132777). For nicking gRNA (ngRNA) cloning, spacer oligos were duplexed and ligated into the Bsmbl-digested pUC19-based hU6-SpCas9 gRNA entry vector BPK1520 (Addgene no. 65777). All pegRNA and ngRNA plasmids were transformed into chemically competent E.coli (XL1-Blue, Agilent). Plasmids used for transfection were midi (PE2) or mini prepped (gRNAs) using the Qiagen midi plus or miniprep kits.

Cell culture

STR-authenticated HEK293T cells (CRL-3216) were grown in Dulbecco's modified Eagle medium (DMEM, Gibco) containing 10% fetal bovine serum (FBS, Gibco) and 1% penicillin-streptomycin antibiotic (Gibco). Cells were kept in a 5% CO₂ incubator at 37 °C. Cells were passaged every 2 to 3 days as cells reached 80% confluency. Cells did not exceed passage 13 for all replicates in this experiment. Mycoplasma testing of the cell culture media took place every 4 weeks with the MycoAlert PLUS mycoplasma detection kit (Lonza) and showed negative results for the duration of this study.

Transfections

HEK293T cells were seeded into 96-well flat-bottom cell culture plates (Corning) for PE treatment at 1.2×10^4 cells/well. Transfections were carried out 18-24 h post-seeding with 30 ng PE2 plasmid, 10 ng pegRNA, and 3.3 ng ngRNA plasmid per transfection (per well, in a 96-well plate). TransIT-X2 (Mirus) was used as the lipofection reagent at 0.3 μ L per transfection.

DNA extraction

Post-transfection (72 h), HEK293T cells were washed using 1x PBS (Corning) and lysed with 43.5 μ L of gDNA lysis buffer (100mM Tris, 200 mM NaCl, 5 mM EDTA, 0.05% SDS), 1.25 μ L of 1 M DTT (Sigma), and 5.25 μ L of Proteinase K per well for 96-well plate experiments. For the untreated cells in 6-well plates, 348 μ L gDNA lysis buffer, 10 μ L DTT, and 42 μ L Proteinase K were added to the cells for lysis. The plates were put into a shaker (500 rpm) at 55°C overnight, and gDNA was extracted using 1.5x paramagnetic beads. Beads with bound gDNA were washed

with 70% ethanol 3 times using a Biomek FX^P Laboratory Automation Workstation (Beckman Coulter) and then eluted in 35 µL 0.1x EB buffer (Qiagen).

Targeted amplicon sequencing

The gDNA concentrations of several samples from different pegRNAs/replicates were measured using the Qubit dsDNA HS Assay Kit (Thermo Fisher). The first PCR was performed to amplify the genomic regions of interest (200-250 bp) using 10-20 ng of gDNA. Primers for PCR1 included Illumina-compatible adapter sequences (Supplementary Data 2). A synergy HT microplate reader (BioTek) was then used at 485/528 nm with the Quantifluor dsDNA quantification system (Promega) to measure the concentration of the first PCR products. PCR products from different genomic amplicons were then pooled and cleaned with 0.7x paramagnetic beads. The second PCR was performed to attach unique barcodes to each amplicon using 50-200 ng of the pooled PCR1 products and barcodes that correspond to Illumina TruSeq CD indexes. The PCR2 products were again cleaned with 0.7x paramagnetic beads and measured with the Quantifluor system before final pooling. The final library was sequenced using an Illumina Miseq (Miseq Reagent Kit v.2; 300 cycles, 2 x 150 bp, paired-end). The FASTQ files were downloaded from BaseSpace (Illumina).

Analysis

Amplicon sequencing data were analyzed with CRISPResso version 2.0.42 with HDR mode.

Downstream analysis was sourced from 'CRISPResso_quantification_of_editing_frequency.txt.'

The frequency of *Desired edit* was determined by taking HDR Unmodified and dividing by

Reads_aligned_all_amplicons and the frequency of *Byproduct* was determined by taking the sum of HDR Modified, Reference Modified, Ambiguous and dividing by Reads_aligned_all_amplicons.

PrimeDesign analysis on ClinVar variants

The ClinVar database was accessed April 8th 2020. Variants were filtered with the following conditions: 1) included a valid GRCh38/hg38 coordinate 2) labeled as *Pathogenic* for the column "ClinicalSignificance" 3) contained a unique identifier determined by the concatenation of columns "Name," "RS# (dbSNP)," and "VariationID." All variants with ambiguous IUPAC code were converted into separate entries with non-ambiguous bases for downstream analysis. Following these steps, the total number of ClinVar variants totaled 69,481. Sequence inputs were formatted for all entries for both the installation and correction of these pathogenic variants. After running PrimeDesign on the ClinVar variants, candidate pegRNA designs were filtered with two criteria: 1) maximum RTT length of 34 nt and 2) minimum homology of 5 nt downstream of the edit. The pegRNAs with *PAM disrupted* annotations have mutations in the dinucleotide GG of the NGG motif, and the ngRNAs with *PE3b*, *PE3b non-seed*, and *PE3b seed* annotations have mismatches anywhere in the protospacer, mismatches outside of PAM-proximal nucleotides 1-10, or mismatches within PAM-proximal nucleotides 1-10, respectively.

Construction of PrimeVar database

The filtered ClinVar variants from the PrimeDesign analysis were used to build a comprehensive database of candidate pegRNA and ngRNA combinations. Prime editing designs are available to

install and correct the pathogenic human genetic variants. PrimeDesign was run with a PBS length range of 10-17 nt, RTT length range of 10-80 nt, and ngRNA distance range of 0-100 bp. All of the pegRNA and ngRNA designs for each variant are stored on PrimeVar (<http://primedesign.pinellolab.org/primevar>).

Code Availability Statement

PrimeDesign¹² was used to design pegRNAs and ngRNAs (<https://github.com/pinellolab/PrimeDesign>). Amplicon sequencing data were analyzed with CRISPResso version 2.0.42 with HDR mode (<https://github.com/pinellolab/CRISPResso2>).

Data Availability Statement

All targeted amplicon sequencing data have been deposited under the BioProject accession number PRJNA688137 at the Sequence Read Archive (SRA), accessible at: <https://www.ncbi.nlm.nih.gov/bioproject/PRJNA688137>. Information related to the clinical variants in the manuscript are available at: <https://www.ncbi.nlm.nih.gov/clinvar/>.

References

1. Anzalone, A. V. *et al.* Search-and-replace genome editing without double-strand breaks or donor DNA. *Nature* **576**, 149–157 (2019).
2. Cong, L. *et al.* Multiplex genome engineering using CRISPR/Cas systems. *Science* **339**, 819-823 (2013).
3. Mali, P. *et al.* RNA-guided human genome engineering via Cas9. *Science* **339**, 823-826 (2013).
4. Hwang, W. *et al.* Efficient genome editing in zebrafish using a CRISPR-Cas system. *Nature Biotechnology* **31**, 227-229 (2013).
5. Komor, A. C. *et al.* Programmable editing of a target base in genomic DNA without double-stranded DNA cleavage. *Nature* **533**, 420-424 (2016).
6. Gaudelli, N. M. *et al.* Programmable base editing of AT to GC in genomic DNA without DNA cleavage. *Nature* **551**, 464-471 (2017).
7. Doench, J. G., Fusi, N., Sullender, M., Hegde, M., Vaimberg, E. W. *et al.* Optimized sgRNA design to maximize activity and minimize off-target effects of CRISPR-Cas9. *Nature Biotechnology* (2016).
8. Landrum, M. J. *et al.* ClinVar: public archive of interpretations of clinically relevant variants. *Nucleic Acids Research* **44**, D862-D868 (2016).
9. Lin, Q., Zong, Y., Xue, C. *et al.* Prime genome editing in rice and wheat. *Nature Biotechnology* (2020).
10. Liu, Y., Li, X., He, S. *et al.* Efficient generation of mouse models with the prime editing system. *Cell Discovery* (2020).

11. Kim, H. K., Yu, G. *et al.* Predicting the efficiency of prime editing guide RNAs in human cells. *Nature Biotechnology* (2020).
12. Hsu, J.Y.H. *et al.* PrimeDesign software for rapid and simplified design of prime editing guide RNAs, pinellolab/PrimeDesign, DOI: 10.5281/zenodo.4429461 (2021).